# Enhancing Endothelial Differentiation of Mesenchymal Stem Cells Derived from Human Turbinates Using Lab-on-a-Chip Technology

**DOI:** 10.3390/medicina61030528

**Published:** 2025-03-18

**Authors:** Do Hyun Kim, Sang Hi Park, Mi-yeon Kwon, Chae-Yoon Lim, Sun Hwa Park, David W. Jang, Se Hwan Hwang, Sung Won Kim

**Affiliations:** 1Department of Otolaryngology-Head and Neck Surgery, College of Medicine, The Catholic University of Korea, Seoul 06591, Republic of Korea; 2Institute of Clinical Medicine Research, College of Medicine, The Catholic University of Korea, Seoul 06591, Republic of Korea; 3Postech-Catholic Biomedical Engineering Institute, College of Medicine, The Catholic University of Korea, Seoul 06591, Republic of Korea; 4Department of Head and Neck Surgery & Communication Sciences, Duke University School of Medicine, Durham, NC 27710, USA

**Keywords:** microfluidics, endothelium, stem cells, culture, humans

## Abstract

*Background and Objectives*: Endothelial cells are essential to various therapeutic strategies for cardiovascular diseases. Developing efficient methods to generate large quantities of well-defined endothelial cells could improve cardiovascular treatment. This study explored the impact of lab-on-a-chip technology on the endothelial differentiation potential of mesenchymal stem cells derived from the human inferior nasal turbinate (hNTSCs). *Materials and Methods*: hNTSCs were isolated from five patients and divided into two groups: an experimental group subjected to lab-on-a-chip technology and a control group following two-dimensional differentiation protocols. The endothelial differentiation capacity of hNTSCs was assessed through histological examination and gene expression analysis. *Results*: Comparative evaluation of traditional differentiation methods and lab-on-a-chip technology indicated that hNTSCs expressed endothelial cell-specific markers, including CD34, KDR, CDH5, and CD31. Notably, CD31, CD34, and CDH5 exhibited significantly elevated expression levels in the lab-on-a-chip system. Additionally, cytokine analysis showed marked increases in IL-1a and IL-8 expression under lab-on-a-chip conditions compared to standard differentiation techniques. *Conclusions*: Lab-on-a-chip technology may enhance the differentiation of hNTSCs into endothelial cells with angiogenic potential, highlighting its promise for future cardiovascular regenerative applications.

## 1. Introduction

Endothelial cells (ECs) are vital components of the cardiovascular system, acting as a crucial interface connecting the heart, blood vessels, and immune system [1]. Dysfunction of ECs is a crucial factor in vascular disorders, including hypertension, cardiovascular diseases, and diabetes [2,3]. Additionally, recent reports show that EC impairment could play a role in conditions not traditionally linked to the vascular system, such as neurodegenerative and chronic inflammatory diseases [4,5].

Establishing a readily available and dependable source of human ECs could aid in the treatment of cardiovascular diseases by enabling researchers to model vascular disorders, identify new therapeutic targets, and develop innovative drugs in vitro. Moreover, generating patient-specific ECs from stem cells holds promise for personalized medicine [6]. ECs are fundamental to the vascularization of bioengineered tissue grafts and whole organs, presenting a potential solution for the replacement of damaged or non-functional tissues [7]. Given their critical roles in multiple therapeutic applications, the generation of large quantities of well-characterized ECs could accelerate the advancement of new cardiovascular treatments.

Mesenchymal stem cells (MSCs) are pluripotent stem cells capable of self-renewal and differentiation into a variety of specialized cell types. These capabilities, together with their immunomodulatory functions, make MSCs a cornerstone of regenerative medicine. In addition to differentiation into various cell types to aid in the repair of damaged tissues at injury sites, MSCs also possess strong paracrine activity. This activity is based on their ability to secrete a variety of bioactive molecules, including growth factors and extracellular vesicles, which have profound effects on surrounding cells. These secreted factors play integral roles in shaping the local microenvironment, promoting tissue healing, and supporting the regeneration of various tissues. Through influences on cellular behaviors, modulation of inflammation, and facilitation of communication between cells, MSCs contribute to the restoration of homeostasis and function in damaged tissues and thus hold tremendous promise for therapeutic applications in treating a range of conditions [8].

However, growing MSCs in vitro poses challenges, including cell death, senescence, and reduced multipotency over time [9]. To overcome these limitations, researchers have explored ways to optimize culture conditions, adjusting factors such as medium type, oxygen concentration, and plate coatings to enhance the viability and functionality of MSCs [10].

Human nasal inferior turbinate stem cells (hNTSCs) have emerged as a promising source of MSCs, particularly due to their accessibility during rhinitis treatments [11,12,13]. These cells demonstrate exceptional resistance to environmental stress, exhibit rapid proliferation, and maintain phenotypic stability across multiple passages. Despite their potential, the impact of lab-on-a-chip technology on the endothelial differentiation of hNTSCs remains unexplored. Therefore, this study introduces an innovative lab-on-a-chip model to examine the effects of this technology on the endothelial differentiation capacity of hNTSCs.

## 2. Materials and Methods

This study was conducted with permission from the Institutional Review Board of the Catholic Medical Center’s Clinical Research Coordination Center (HC15TISI0022). The research adhered to the ethical principles outlined in the Declaration of Helsinki, ensuring that all aspects of the study were ethically sound and compliant with international standards. Written informed consent was obtained from all participants prior to cell collection, ensuring that they were fully informed of the nature of the study, potential risks, and their rights throughout the study.

### 2.1. Cell Isolation and Culture

Inferior turbinate tissue was collected from five adults (aged 20 years or older) who underwent partial turbinectomy. These patients were then assigned to one of two groups: the study group, which utilized lab-on-a-chip differentiation methods, or the control group, which followed conventional differentiation techniques. The selection criteria for participants were carefully established to ensure the reliability of the results. Individuals diagnosed with antrochoanal polyps, nasal polyposis, or congenital immunological disorders were excluded to eliminate potential variables that could affect the outcomes. These exclusion criteria were designed to ensure that only patients with no underlying conditions affecting their nasal tissues or immune responses were included.

The process of isolating hNTSCs from the collected turbinate tissues involved multiple steps to maintain sterility and optimal cell viability. First, the tissue samples were subjected to extensive washing to remove contaminants and minimize the risk of microbial infection. Specifically, the samples were rinsed three to five times using an antibiotic–antimycotic solution (Gibco, Gaithersburg, MD, USA) to eliminate potential bacterial or fungal contaminants. After this initial decontamination step, the tissues underwent two additional washes with phosphate-buffered saline (PBS) to ensure the removal of residual antibiotic solution and any remaining impurities. Following the washing process, the tissues were carefully cut into small fragments, each measuring approximately 1 mm^3^. This fragmentation process was conducted to facilitate the subsequent attachment and expansion of stem cells in culture. The prepared tissue fragments were transferred into culture dishes that had been pre-coated with CELLstart Substrate (Gibco) to optimize cell adherence and proliferation. To ensure stability and prevent displacement during the early stages of culturing, a sterilized glass coverslip was placed over the tissue fragments. Once the tissue fragments were securely positioned, a culture medium was introduced to support stem cell growth. The culture medium used was StemPro MSC SFM XenoFree (Gibco), a specialized serum-free medium designed for MSC expansion. Additionally, the medium was supplemented with 200 mM L-glutamine (Gibco) to enhance cell viability and metabolic activity. The culture medium was changed every other day to maintain a nutrient-rich environment and remove metabolic waste.

After 3 weeks, any remaining floating tissue fragments that had not adhered were removed via washing with PBS to ensure a clean culture environment. At this stage, adherent hNTSCs were enzymatically detached with TrypLE Select Enzyme 10× (Gibco). This process allowed for the efficient collection of viable cells without causing excessive damage.

To further evaluate the characteristics of the cultured hNTSCs, cells at four different passages were subjected to immunophenotypic analysis. This assessment was conducted to identify any potential alterations to cell surface markers caused by prolonged culturing and exposure to specific culture media. Additionally, the cells were examined for their potential to undergo endothelial differentiation, providing insight into their functional capabilities and suitability for regenerative applications.

### 2.2. Characterization of hNTSCs Through Cell Surface Marker Analysis

Flow cytometry was used to assess the expression of specific cell surface markers in hNTSCs, providing insights into their immunophenotypic characteristics. To prepare the cells for analysis, they were suspended in PBS at a density of 1 × 10^5^ cells/mL. This standardized concentration ensured consistency across samples and facilitated accurate comparative analysis. To eliminate any residual culture medium, unbound proteins, or debris that could interfere with antibody binding, the cells were subjected to three sequential washes with PBS. This washing process helped to improve the specificity and reliability of the staining procedure. Following preparation, the cells were incubated for 1 h with primary monoclonal antibodies targeting key surface markers. These markers were selected based on their relevance to characterizing MSCs and distinguishing MSCs from hematopoietic or immune cell populations. The antibodies were applied at saturating concentrations to ensure maximal binding and to prevent variability in signal intensity. All anti-human cluster of differentiation (CD) antibodies used in this study were obtained from BD Biosciences (San Jose, CA, USA), a reputable supplier of high-quality flow cytometry reagents, to ensure the consistency and reliability of the results. After primary antibody incubation, the cells were washed an additional three times with buffer to remove any unbound antibodies and reduce background fluorescence. This step was critical, as it ensured that the fluorescence signals detected during flow cytometry represented bound antibodies rather than non-specific interactions. The washed cells were then subjected to centrifugation at 400× *g* for 5 min to produce a pellet of concentrated cells for further processing. The pellet was gently resuspended in ice-cold PBS to maintain cellular integrity and prevent degradation. To promote secondary antibody binding and enhance fluorescence signal detection, the cells were incubated with secondary antibodies for 30 min in the dark at 4 °C. This incubation was conducted in the dark to prevent photobleaching, which could otherwise diminish fluorescence intensity and thus compromise data accuracy. This step ensured that the labeled antibodies remained intact and provided a reliable fluorescence signal for subsequent analysis. Flow cytometry was performed using a FACSCalibur flow cytometer (BD), which enabled the detection and quantification of fluorescence signals corresponding to each surface marker, providing a detailed profile of the hNTSCs’ immunophenotypic characteristics. The acquired data were processed and analyzed using CellQuest (Ver. 6.0) software (BD), which facilitated graphical representation and statistical evaluation of marker expression. Through this rigorous analytical approach, the present study aimed to comprehensively assess the surface marker profile of hNTSCs and confirm their MSC properties.

### 2.3. Endothelial Differentiation of hNTSCs Using a Lab-on-a-Chip Model

A human-specific lab-on-a-chip system was developed to replicate both endothelial and stromal conditions. The device consisted of three main components: an upper chamber, a lower chamber, and a polycaprolactone electrospun nanofibrous membrane between the two chambers. The device was designed for easy cultivation, fitting into a 12-well plate (Figure 1). The membrane was synthesized through dissolution in a solvent containing chloroform and dimethylformamide at a 9:1 ratio, and then electrospun to produce nanofibers with a diameter of 200 nm (total thickness, 200 µm). Electrospinning was performed with an inner diameter of 0.4 mm at room temperature under conditions of 2 mL/hour flow rate and 18 kV applied voltage. Prior to seeding, the chip (PETE membrane, 0.4 µm pore size, Sterlitech, Auburn, WA, USA) was pre-coated with a mixture of type-IV collagen at 60 µg/mL (Sigma-Aldrich, St. Louis, MO, USA) and Matrigel at 1/80 dilution (Corning, Acton, MA, USA). hNTSCs were seeded onto the membrane (4 × 10^4^ cells/well).

hNTSCs were cultured in a growth medium to enhance their viability prior to seeding. The medium consisted of low-glucose Dulbecco’s Modified Eagle Medium (DMEM) containing 10% fetal bovine serum (FBS), 2 mM GlutaMAX, and 5 μg/mL gentamicin. The cells were incubated in this medium for 1 h and then seeded onto the membrane of the lab-on-a-chip system at a density of 4 × 10^4^ cells/well. During the initial phase, the cells were cultured under submerged conditions in a seeding medium containing low-glucose DMEM supplemented with 10% FBS, 2 mM GlutaMAX, and 5 µg/mL gentamicin. To support cell attachment and proliferation, cells were placed in an incubator at 37 °C under 5% CO_2_ for 2 days. Following this incubation period, differentiation was induced by transitioning the cultures to air–liquid interface (ALI) conditions using a specialized differentiation medium. This medium consisted of EGM-2 BulletKit medium (CC-3162, Lonza, Walkersville, MD, USA) supplemented with two recombinant proteins: 50 ng/mL vascular endothelial growth factor (VEGF, Peprotech, Cranbury, NJ, USA) and 25 ng/mL basic fibroblast growth factor (FGF-basic, Peprotech, Rocky Hill, NJ, USA). Differentiation was conducted over a 21-day period, during which the cultures were maintained under ALI conditions to facilitate mucociliary differentiation. At the end of this period, the differentiation status of the cultures was evaluated using lineage-specific biological stains to confirm the development of distinct epithelial and stromal characteristics, ensuring that the cells acquired the appropriate phenotypes.

To establish a baseline for comparison, a control group was incorporated into the study. In this group, hNTSCs were cultured under the conditions described above, but on a 6-well confocal plate (SPL Life Sciences, Seoul, Republic of Korea) or a 12-well culture plate rather than the lab-on-a-chip system. The control cells were seeded at the same density and maintained in the same differentiation medium under ALI conditions. The inclusion of this control group enabled direct evaluation of the impact of the microengineered lab-on-a-chip system on cellular differentiation through comparison with a conventional culturing approach.

To further assess endothelial differentiation, histological examination and immunofluorescence staining were conducted on the differentiated cells. The stained samples were observed under an inverted microscope to identify morphological and structural changes indicative of endothelial differentiation. Additionally, gene expression patterns associated with differentiation were evaluated using reverse transcription polymerase chain reaction (RT-PCR).

### 2.4. Tube Formation Assay

Differentiated hNTSCs, which had been cultured at a density of 1.2 × 10^5^ cells for 21 days, were seeded into 24-well culture plates coated with Matrigel. For immunofluorescence staining, the cells were labeled with 8 µg/mL Calcein AM (Corning, Corning, NY, USA) and incubated for 30 min at 37 °C under an atmosphere of 5% CO_2_. Images of the labeled cells were captured using a microscope.

### 2.5. Cytokine Assays

hNTSCs were seeded into 12-well plates (8 × 10^4^ cells/well) and allowed to attach overnight. For analysis, the conditioned medium was sampled on days 0, 7, and 14 of culturing from both the microchip and control groups. Cytokines and chemokines (interleukin [IL]-1α, IL-1β, IL-5, IL-6, IL-8, IL-10; monocyte chemoattractant protein 1 [MCP-1]; transforming growth factor beta 1 [TGF-β1]; C-C motif ligand 5 [CCL5, also known as RANTES]; tumor necrosis factor α [TNF-α]; interferon γ [IFN-γ]; and granulocyte-macrophage colony-stimulating factor [GM-CSF]) were analyzed using the MILLIPLEX MAP human cytokine/chemokine multiplex immunoassay (Millipore, Billerica, MA, USA). To ensure the reliability and reproducibility of the results, experiments were performed independently on at least three separate occasions using different MSC donor pools to mitigate potential biological variability.

### 2.6. RT-PCR of hNTSCs

Total RNA was extracted from ECs cultured within the lab-on-a-chip system using a QIAzol lysis reagent (QIAGEN, Valencia, CA, USA). Briefly, the membrane located between the upper and lower parts of the chip device was thoroughly rinsed with cold PBS. After the removal of the membrane, it was placed in a tube and homogenized using an Intelli-Mixer RM-2M rotary mixer (ELMI, Riga, Latvia) with 300 μL of QIAzol dissolution reagent. Following homogenization, the cellular material was collected, and phenol/chloroform extraction was performed. Isolated RNA was quantified using a BioPhotometer D30 instrument (Eppendorf, Hamburg, Germany), and 500 ng of purified RNA was reverse transcribed into complementary DNA (cDNA) using CellScript cDNA Synesis Master Mix (CellSafe, Suwon, Republic of Korea), during which a genomic DNA removal step was performed. For comparative analysis, cells were also cultured under respiratory differentiation conditions using conventional two-dimensional (2D) culturing techniques. After culturing, 350 µL of QIAzol lysis reagent was added to the cell cultures, and the lysates were processed using previously described methods. Gene expression analysis was performed via RT-PCR amplification and relative quantification of essential endothelial markers such as CD-31, CD-34, von Willebrand factor (VWF), cadherin 5 (CDH5, also known as vascular endothelial [VE]-cadherin), and vascular endothelial growth factor receptor-2 (VEGFR-2, also known as KDR). TaqMan gene expression assays (Applied Biosystems, Foster City, CA, USA) were performed using the LightCycler 480 PCR system (Roche, Mannheim, Germany) (Table 1). Assays were designed to maintain consistent amplification efficiency, and relative quantification was conducted using the delta cycle threshold (ΔCt) method. Each reaction was carried out in triplicate in a total volume of 10 μL, utilizing TaqMan Probe Master Mix (Roche) and 100 ng of cDNA per reaction. Glyceraldehyde 3-phosphate dehydrogenase (GAPDH) was used as an endogenous control to normalize the data. Data analysis was performed using the LightCycler 480 software ver. 1.2 (Roche).

### 2.7. Histological Examination and Immunofluorescent Staining

After collection and fixation, the samples underwent dehydration through a series of increasingly concentrated ethanol solutions, followed by clearing in xylene. Subsequently, the samples were embedded in paraffin for sectioning. The resulting slides were left to dry overnight to ensure that the sections adhered securely, and then stored at 4 °C until further processing. To assess general tissue morphology, the samples were stained using hematoxylin and eosin, allowing for a comprehensive analysis of tissue architecture.

To stain hNTSCs, the paraffin sections were deparaffinized, followed by rehydration with ethanol solutions to prepare the tissues for subsequent analysis. To expose the antibody-binding sites, antigen retrieval was conducted using a proteinase K solution (Abcam, Cambridge, UK). Then, the samples were incubated with primary antibodies targeting key endothelial markers, including anti-CD31, anti-CD34, anti-VE-cadherin, anti-VEGFR-2, and anti-VWF sourced from either rabbit or mouse. The sections were then incubated with secondary antibodies conjugated to fluorescent dyes, namely Alexa Fluor 546 (green) or Alexa Fluor 488 (red), corresponding to the species of the primary antibodies (goat anti-rabbit IgG or goat anti-mouse IgG, respectively, diluted at 1:1000). This staining method enabled the visualization of endothelial markers within the tissue samples, providing valuable insights into cellular composition and the endothelial differentiation processes. Fluorescent quantification was measured using the ImageJ (https://imagej.net/, accessed on 3 March 2025) program.

### 2.8. Statistical Analysis

Statistical analysis was conducted using R software version 4.4.2 (R Foundation for Statistical Computing, Vienna, Austria), which is widely used for data analysis and statistical computing. To assess the differences among the groups, *t*-tests and one-way analysis of variance (ANOVA) were employed, depending on the nature of the data and the specific comparisons being made. The *t*-test was used for comparisons of two groups, while one-way ANOVA was used to assess differences across more than two groups. A *p*-value of less than 0.05 was considered to indicate statistical significance. All statistical tests were performed with the assumption of normality and equal variances, where applicable.

## 3. Results

### 3.1. Characterization of hNTSCs

The hNTSC population expressed typical MSC markers (CD29, CD73, and CD90), and was negative for hematopoietic markers (CD14, CD34, and human leukocyte antigen-DR isotype [HLA-DR]), consistent with the expected phenotype of MSCs (Figure 2 and Figure 3).

### 3.2. Cytokine and Chemokine Secretion Patterns of hNTSCs

We investigated the secretion profiles of various immunomodulatory cytokines and chemokines, including GM-CSF, IFN-γ, IL-1α, IL-1β, IL-5, IL-10, RANTES, TNF-α, IL-6, IL-8, MCP-1, and TGF-β1 (Figure 4). Among these, GM-CSF, IL-1α, IL-10, RANTES, TNF-α, IL-6, IL-8, MCP-1, and TGF-β1 were detected in measurable quantities in hNTSC supernatants from both experimental groups, with average concentrations exceeding 1 pg/mL. However, significant differences in secretion levels were found between the lab-on-a-chip and control groups for some cytokines and chemokines. The lab-on-a-chip group exhibited increased secretion of IL-1α and IL-8 compared to the control group. Notably, IL-8 levels were approximately four times higher in the lab-on-a-chip group, while RANTES secretion was about four times lower in the lab-on-a-chip group compared to the control (Figure 4). These results suggest that culture methods can influence the immunological response of hNTSCs.

### 3.3. Endothelial Differentiation Potential of hNTSCs

To assess the effectiveness of differentiation protocols, hNTSCs were exposed to endothelial differentiation conditions used in standard 2D culturing for 3 weeks. The formation of cord-like structures was examined through phase-contrast microscopy and immunofluorescence staining. Both methods showed a network of tubes, loops, mesh structures, and nodes. In contrast, hNTSCs cultured in a non-differentiation medium did not form tube networks (Figure 5). Importantly, after 3 weeks of differentiation, both the total cell count and the number of cells exhibiting immunofluorescence staining were higher than at the 2-week mark (Figure 6 and Appendix A). These results suggest that 3 weeks is an ideal period under differentiation conditions for endothelial differentiation. To determine the capacity of expanded hNTSCs from both groups to differentiate into ECs, histological methods were used to examine endothelial markers, including CD31, CD34, VE-cadherin, VEGFR-2, and VWF (Figure 7 and Appendix A). Immunofluorescence staining confirmed that endothelial differentiation conditions led to the expression of these markers in cell monolayers originating from both groups. Additionally, RT-PCR was employed for quantitative analysis of gene expression related to differentiation. Cells cultured in differentiation medium showed elevated mRNA levels of EC-specific markers, including CD34, KDR, CDH5 (VE-cadherin), and CD31. Specifically, CD31, CD34, and CDH5 exhibited significant upregulation in the lab-on-a-chip group compared to the control group (Figure 8).

## 4. Discussion

MSCs have received considerable research attention due to their potential applications in regenerative medicine and cell-based therapies. In addition, their capacity to influence immune responses and release bioactive factors positions them as strong candidates for the treatment of inflammatory and degenerative diseases. However, successfully applying laboratory findings to the development of in vivo therapies remains challenging, as animal studies often reveal inconsistencies between experimental data and clinical results.

A major challenge to the clinical application of MSCs is the difficulty associated with their in vitro expansion and differentiation. During culturing, MSCs frequently encounter several obstacles, including reduced proliferation rates, cellular senescence, and diminished multipotency, all of which can compromise their therapeutic efficacy [14]. The successful cultivation of MSCs is influenced by a variety of factors, including the composition of the culture medium, oxygen concentration, and properties of the extracellular matrix. Thus, optimizing these parameters is essential to preserving the quality, viability, and functional potential of MSCs for clinical applications.

hNTSCs have emerged as a promising source of MSCs, largely due to their accessibility during routine surgical procedures for chronic rhinitis. These cells exhibit remarkable resistance to environmental stressors and robust proliferative capacity, allowing them to maintain their characteristics across multiple passages. Furthermore, hNTSCs possess inherent advantages over other MSC sources, such as ease of isolation and potential for large-scale expansion. Despite the growing interest in hNTSCs as MSC-like cells, research remains limited regarding how various culture conditions influence their differentiation potential and functional properties. In particular, the impact of advanced culture models, such as microfluidic lab-on-a-chip systems, on hNTSC behavior has yet to be fully explored, highlighting the need for further investigation in this area.

For endothelial differentiation, cell culturing has typically been conducted using 2D culture flasks and media to form a monolayer of cells. However, this conventional 2D culturing method does not provide appropriate microenvironmental conditions. Therefore, the development of a lab-on-a-chip based on a three-dimensional culturing system was undertaken to provide a microenvironment similar to the extracellular matrix. In our previous report, we found that this lab-on-a-chip system influenced the characteristics of hTNSCs in terms of epithelial differentiation and immunomodulation, suggesting that this chip may provide microenvironmental conditions appropriate for hTNSCs [15]. Based on this background, we assumed that this chip could also affect the endothelial differentiation of hTNSCs.

The present study demonstrates that hNTSCs express EC-specific markers, such as CD34, KDR, CDH5, and CD31 (platelet EC adhesion molecule), when differentiated using the lab-on-a-chip method, in contrast to traditional differentiation techniques. Cells cultured in the lab-on-a-chip system showed significantly higher levels of these markers, with notable differences in CD31, CD34, and CDH5. Interestingly, hTNSCs on the chip tended to differentiate into ECs more readily in both undifferentiation media and differentiation media compared to 2D culture conditions. Although this study does not clarify the basic mechanism of endothelial differentiation in the lab-on-a-chip, the three-dimensional environment of the chip may influence the characteristics of hTNSCs. Further research should be conducted to identify the mechanisms underlying differentiation in this chip system. Cytokine analysis revealed marked increases in IL-1α and IL-8 in the lab-on-a-chip group. IL-1α plays a role in angiogenesis, influencing vascular proliferation, migration, and tube formation [16]. Additionally, IL-1α is linked to high secretion of IL-8, which promotes EC survival, proliferation, matrix metalloproteinase production, and angiogenesis [17,18]. In contrast, RANTES expression exhibited the opposite trend. RANTES, a chemokine associated with inflammation, contributes to endothelial dysfunction and promotes vascular inflammation by activating nuclear factor kappa B and nicotinamide adenine dinucleotide phosphate oxidase 1 in ECs [19]. These results suggest that the lab-on-a-chip system significantly improves endothelial differentiation of hNTSCs compared to traditional 2D culture methods.

The microfluidic device developed in this study provides a platform to enhance the endothelial differentiation potential of hNTSCs. This system is designed to replicate the endothelial environment, offering more accurate context for studies of hNTSC behavior. This approach aims to address some of the limitations of conventional 2D culture systems and to elucidate hNTSC differentiation and functionality.

Recently, stem cell research has increasingly focused on applying endothelial differentiation to clinical settings, particularly for vascular regeneration, personalized disease modeling, and drug development. One strategy for vascular regeneration involves the injection of autologous stem cell-derived ECs into ischemic tissues, aiming to stimulate neovascularization and restore blood flow, thereby improving tissue function. This approach may offer some therapeutic benefits for patients with conditions including refractory angina [20,21], ischemic heart failure [22], non-ischemic dilated cardiomyopathy [23], and critical limb ischemia [24], and has shown potential in enhancing cardiac perfusion and ventricular function [25]. In terms of personalized disease modeling and drug development, the creation of accurate models for vascular diseases such as atherosclerosis, diabetes, and hypertension is challenging due to their complex, multifactorial nature and the interplay between genetic and environmental factors. Integration of stem cell-derived ECs into such models could aid in the prediction of patient-specific disease susceptibility and treatment responses [26].

While the in vitro model of EC differentiation demonstrated a strong functional cytokine response, other physiological components, such as immune cells, were absent. This limitation highlights a large gap in the current organ-on-a-chip platform, and further research is needed to improve the physiological relevance of this technique. The small sample size of this study is an additional constraint. Despite these limitations, the model holds considerable promise for supporting vascular growth in ischemic tissues and tackling endothelial dysfunction in various diseases. The potential availability of autologous human ECs holds major therapeutic promise.

## 5. Conclusions

hNTSCs may serve as an alternative source of endothelial progenitors for clinical applications, such as tissue regeneration or the vascularization of artificial organs. Moreover, in vitro differentiation of hNTSCs using the lab-on-a-chip model may provide a valuable system for investigating the mechanisms underlying endothelial differentiation.

## Figures and Tables

**Figure 1 medicina-61-00528-f001:**
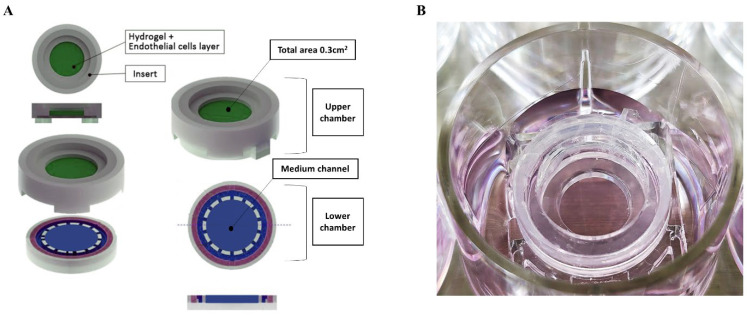
A schematic diagram of the respiratory mucosa-on-a-chip model developed in this study (**A**), and the device placed in the well of a 12-well plate (**B**).

**Figure 2 medicina-61-00528-f002:**
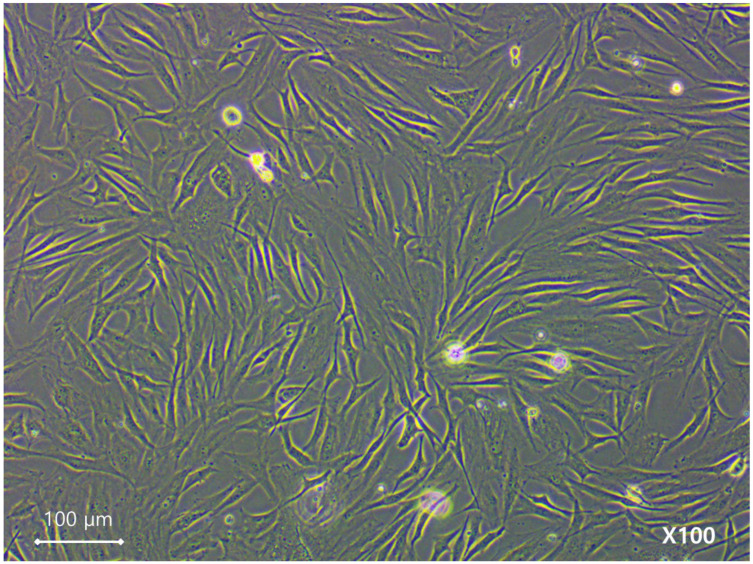
Cell morphology following initial explant cultivation. Cells in both groups attached to the culture vessel and exhibited similar elongated, fibroblast-like shapes (magnification, 100×).

**Figure 3 medicina-61-00528-f003:**
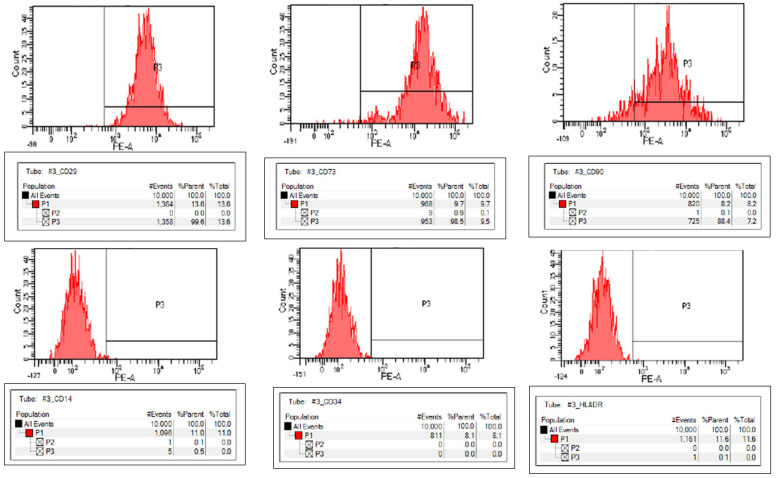
Flow cytometry analysis of human turbinate mesenchymal stromal cells (hNTSCs). hNTSC (from a single donor) expressed CD29, CD73, and CD90, while lacking expression of CD14, CD34, and HLA-DR.

**Figure 4 medicina-61-00528-f004:**
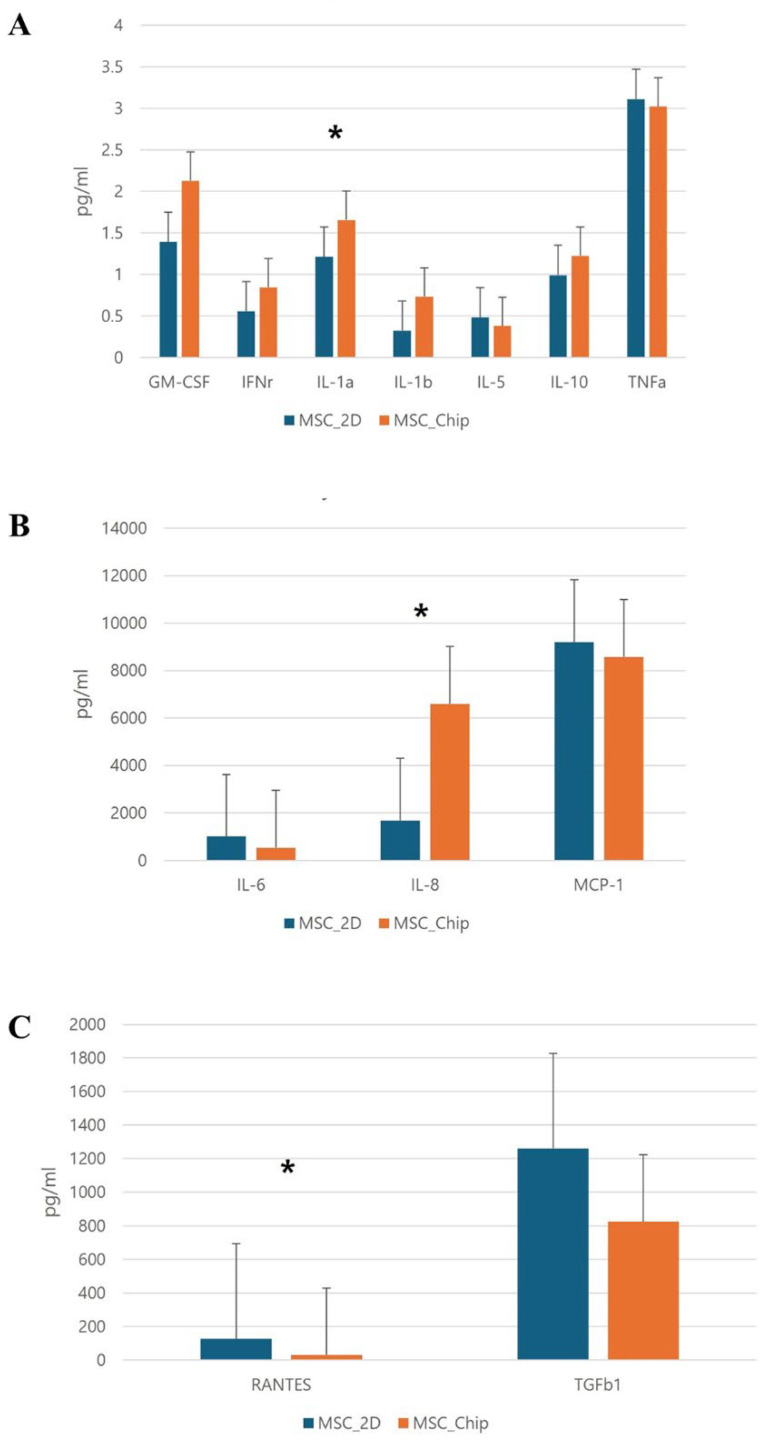
Evaluation of cytokine and chemokine production by hNTScs grown using lab-on-a-chip and traditional culture methods. GM-CSF, IFN-γ, IL-1α, IL-1β, IL-5, IL-10, and TNF-α (**A**); IL-6, IL-8, and MCP-1 (**B**); RANTES and TGF-β1 (**C**) (error bars represent ± standard error [SE], *: *p* < 0.05).

**Figure 5 medicina-61-00528-f005:**
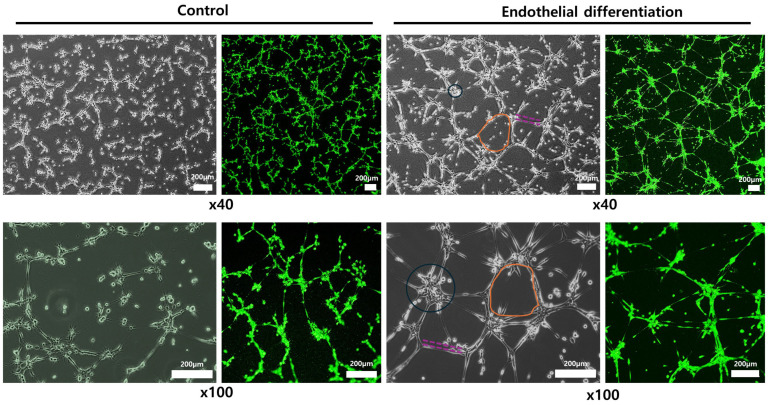
Endothelial differentiation potential of hNTSCs. hNTSCs were cultivated under non-differentiation (**left**) and endothelial differentiation (**right**) conditions in conventional 2D culture for 3 weeks. Under differentiation conditions, the formation of cord-like structures was visualized through phase-contrast microscopy and immunofluorescence staining. Upon more detailed morphological examination, tubes (dotted purple lines), loops/meshes (orange circles), and nodes (blue circles) in the tube network could be identified in both images (from a single donor).

**Figure 6 medicina-61-00528-f006:**
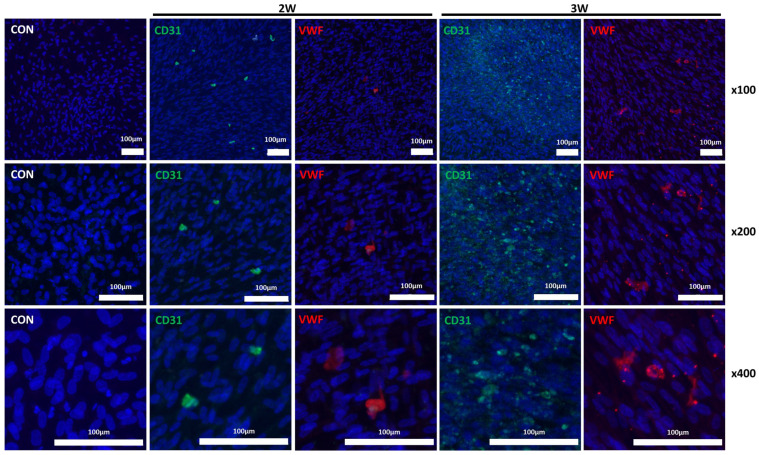
Endothelial differentiation over time. Under differentiation conditions, hNTSCs expressed the endothelial markers CD31 (green) and VWF (red), which were not expressed by the control group, as assessed via immunofluorescence staining (4′,6-diamidino-2-phenylindole [DAPI]; blue) of cells from a single donor.

**Figure 7 medicina-61-00528-f007:**
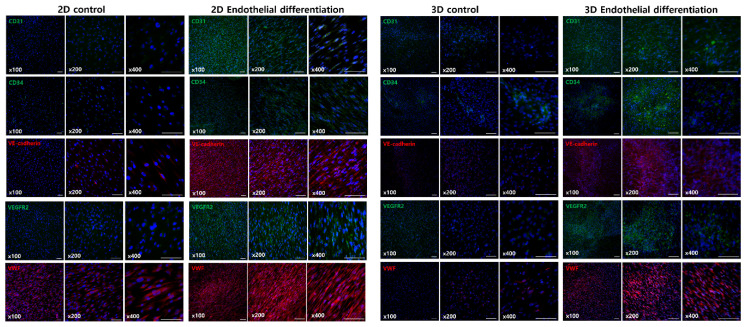
Immunofluorescence staining of CD31, CD34, VEGFR-2, VE-cadherin, and VWF. hNTSCs differentiated under lab-on-a-chip conditions included more immunofluorescence-stained cells compared to the control group. This pattern was similar to the result obtained using conventional differentiation methods with cells from a single donor.

**Figure 8 medicina-61-00528-f008:**
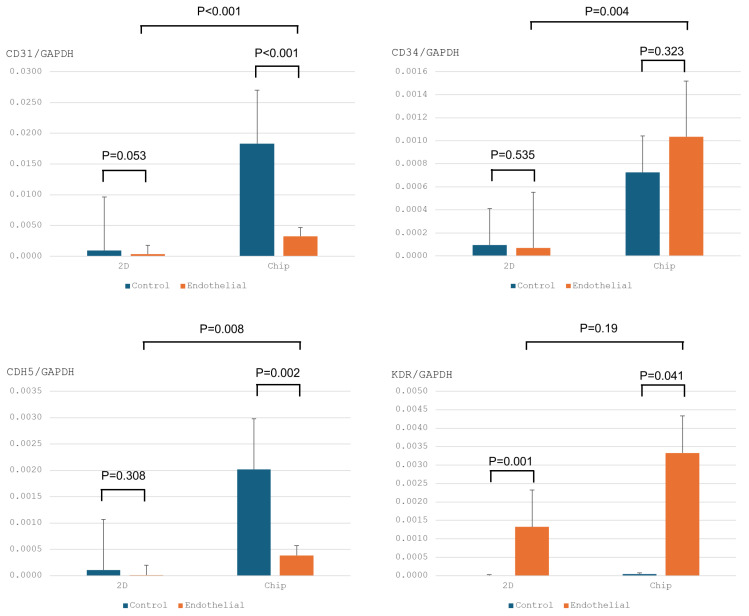
RT-PCR analysis after 3 weeks of induction in endothelial differentiation media. hNTSCs expressed the endothelial markers CD34, KDR, CDH5, and CD31, as observed through real-time RT-PCR analysis (bars indicate ± SE).

**Table 1 medicina-61-00528-t001:** Gene expression assays used for RT-PCR analysis to confirm endothelial differentiation.

Gene	Abbreviation	Reference Sequence	Assay Number
Cluster of differentiation 31	CD31	NM_000442	Hs01065279_m1
Cluster of differentiation 34	CD34	NM_001773	Hs02576480_m1
Von Willebrand factor	VWF	NM_000552	Hs01109446_m1
Cadherin 5	CDH5	NM_001795	Hs00901465_m1
Vascular endothelial growth factor receptor 2	VEGFR-2	NM_002253	Hs00911700_m1
Glyceraldehyde-3-phosphate	GAPDH	NM_002046	Hs99999905_m1

## Data Availability

Information excluding patient-related information may be provided upon request to the corresponding authors.

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
