# Peer review of "Enhancing Endothelial Differentiation of Mesenchymal Stem Cells Derived from Human Turbinates Using Lab-on-a-Chip Technology"

_medicina, 2025, doi:10.3390/medicina61030528_

Round 1

Reviewer 1 Report

Comments and Suggestions for Authors

This study aimed to demonstrate that the Lab-on-a-Chip technique enhances the ability of human nasal inferior turbinate stem cells (hNTSCs) to differentiate into endothelial cells. The authors previously showed that hNTSCs are a valuable source of mesenchymal stem cells (MSCs) due to their high proliferation rate, resistance to environmental stress, and stable phenotypic integrity. In this study, the authors applied the Lab-on-a-Chip technique to demonstrate improved differentiation potentials of hNTSCs to endothelial cells by showing marker expressions using various experimental method. Despite novel concept, the study lacks the experimental rigor, making it difficult to support the authors’ conclusion. In addition, Lab-on-a-Chip system used in this study should be compared to the one presented in the authors’ prior publication in Medicina (Kim et al, 2024), which was not included in the reference list.

Major points

  1. The rigor of this study is weak.

- lack of biological replicates (information) (all Figures)

- lack of statistics (Figure 3)

- lack of quantification (Figures 6 and 7)

- lack of proper controls (negative controls were missing in Figures 7 and 8)

  1. Chip system used in this study appears to be the same as the one described in the authors’ previous report (Kim et al, Medicina, 2024), which was missing from the references. If it is the same system, the study should describe the differentiation conditions that distinguish the different cell types produced.

  1. The manuscript should be proofread.

- page 9: The description regarding IL-8 is confusing.

- page 11: The description regarding KDR expression is confusing.

Comments on the Quality of English Language

# The manuscript should be proofread. (Same as 3.)

- page 9: The description regarding IL-8 is confusing.

- page 11: The description regarding KDR expression is confusing.

Author Response

Comments and Suggestions for Authors

Major points

  1. The rigor of this study is weak.

- lack of biological replicates (information) (all Figures)

Figure 2, 3, 5, 6, and 7 stemmed from 1 donor’s sample (it does not mean the same person). As we know, the histologic findings were checked in individual samples. However, for the submission and publication, the authors chose the best quality figures from the single person although the histologic finding from the enrolled patients showed the same results.

â—Ž Reply:

As the comment of the reviewer, we added the biological replicates (information) in the figure legends.

- lack of statistics (Figure 3)

â—Ž Reply:

As mentioned in the response above, Figure 3 is based on data from a single individual; therefore, statistical analysis was not applied.

- lack of quantification (Figures 6 and 7)

â—Ž Reply:

We quantified fluorescence using the ImageJ program and added it as Supplementary Figure 1 and Supplementary Figure 2.

- lack of proper controls (negative controls were missing in Figures 7 and 8)

 Figure 7 showed the negative control as 2D control or 3D control.

In figure 8. Previous version did not show the negative control.

â—Ž Reply:

In accordance with the reviewer's comment, we have revised the version to align with the one that includes the control.

  1. Chip system used in this study appears to be the same as the one described in the authors’ previous report (Kim et al, Medicina, 2024), which was missing from the references. If it is the same system, the study should describe the differentiation conditions that distinguish the different cell types produced.

â—Ž Reply:

According to the comment of reviewer, we added the missed the reference (Ref. 15).

In the Methods section, we previously described the differentiation conditions (media combined with the lap-on-a-chip system). However, in the revised manuscript, we have provided a more detailed explanation of the application of the lap-on-a-chip system for endothelial differentiation. In our previous report, we demonstrated that the chip system was designed for air-fluid interface culture and that it successfully facilitated the differentiation of hNTSCs into respiratory mucosa. For endothelial differentiation, traditional cell culture has typically been conducted using two-dimensional (2D) plastics or glassware, allowing for the formation of a monolayer of cells in media. However, conventional 2D culture systems are limited in their ability to provide adequate microenvironmental conditions, particularly concerning cell–cell and cell–extracellular matrix (ECM) interactions. Consequently, the development of three-dimensional (3D) culture systems was initiated to better replicate relevant microenvironments. To enhance cell–cell adhesion and interactions with the substrate, various biomaterial-based 3D scaffolds and spheroids—a scaffold-free 3D technique—have been developed to mimic the ECM.

Recently, advancements in three-dimensional culture systems, particularly those incorporating 3D printing-based chip fabrication, have addressed the demand for rapid prototyping, low-volume production, controlled complex microstructures, and tissue mimicry. Notably, the automation of these systems reduces fabrication steps and time, eliminates the need for assembly and bonding, and lowers manufacturing costs. Building upon this background, our previous study demonstrated that the lap-on-a-chip system influenced the characteristics of hNTSCs during epithelial differentiation and immunomodulation, suggesting that the chip may provide a unique microenvironmental condition for hNTSCs.

We hypothesized that the specific properties of this chip could also exert an effect on the endothelial differentiation of hNTSCs. Interestingly, in the quantitative RT-PCR analysis, the chip appeared to promote the differentiation of hNTSCs into endothelial cells under both control and differentiation media, showing greater efficacy compared to conventional 2D culture systems. Although the current study does not elucidate the fundamental mechanisms underlying endothelial differentiation in the lap-on-a-chip system, it is possible that the three-dimensional environment provided by the chip influences the characteristics of hNTSCs. In the next phase, further studies will be conducted to investigate the mechanisms of differentiation within this system.

The aforementioned content has been incorporated into the Discussion section.

  1. The manuscript should be proofread.

â—Ž Reply:

According to the comments, we performed English proofreading again.

The English in this document has been checked by at least two professional editors, both native speakers of English. For a certificate, please see:

  http://www.textcheck.com/certificate/vdzvto

- page 9: The description regarding IL-8 is confusing.

â—Ž Reply:

There were typos that misled the data understanding. As the comment of reviewer, we corrected the mistakenly written contents.

- page 11: The description regarding KDR expression is confusing.

the lab-on-a-chip group exhibited higher expression levels of all endothelial markers compared to the conventional differentiation group. However, in view of statistical significance, the increased level of KDR was not significantly higher than that of conventional culture. By contrast, the other markers (CD31, CD34, and CDH5) in lab-on-a-chip group showed significant upregulation in the lab-on-a-chip group compared to the control group.

â—Ž Reply:

In previous version, this contents could make the reader misunderstand or confuse, so the sentence,” Notably, the lab-on-a-chip group exhibited higher expression levels of all endothelial markers compared to the conventional differentiation group.”, was deleted.

Comments on the Quality of English Language

# The manuscript should be proofread. (Same as 3.)

â—Ž Reply:

According to the comments, we performed English proofreading again.

The English in this document has been checked by at least two professional editors, both native speakers of English. For a certificate, please see:

  http://www.textcheck.com/certificate/vdzvto

- page 9: The description regarding IL-8 is confusing.

â—Ž Reply:

There were typos that misled the data understanding. As the comment of reviewer, we corrected the mistakenly written contents.

- page 11: The description regarding KDR expression is confusing.

the lab-on-a-chip group exhibited higher expression levels of all endothelial markers compared to the conventional differentiation group. However, in view of statistical significance, the increased level of KDR was not significantly higher than that of conventional culture. By contrast, the other markers (CD31, CD34, and CDH5) in lab-on-a-chip group showed significant upregulation in the lab-on-a-chip group compared to the control group.

â—Ž Reply:

In previous version, this contents could make the reader misunderstand or confuse, so the sentence,” Notably, the lab-on-a-chip group exhibited higher expression levels of all endothelial markers compared to the conventional differentiation group.”, was deleted.

Reviewer 2 Report

Comments and Suggestions for Authors

Thank you for an interesting paper.

Author Response

Comment on Figure 4:

  1. Why the cytokines in panel A has such low concentrations in general, in contrast to panel B and C? If the concentration is too low, is it really meaningful to compare between two groups?

â—Ž Reply:

In accordance with the reviewer's comment, the cytokine concentrations in Panel A were low, ranging from 0.3 to 5 pg/ml. This low concentration may be attributed to either the dilution of the supernatant or minimal secretion from the cells. 

However, a previous study compared the cytokine levels of human umbilical cord (UC) and bone marrow (BM) MSCs under various environmental conditions. In that study, certain cytokines—IL-10, IL-13, TNF-α, IFN-γ, eotaxin, IL-9, IL-1β, IL-4, and MIP-1β—were secreted at low concentrations (<10 pg/ml per million cells) by BM-MSCs, while their levels in conditioned media from UC-MSCs ranged from undetectable to <2 pg/ml (*Cytokine Activation Reveals Tissue-Imprinted Gene Profiles of Mesenchymal Stromal Cells. Front. Immunol. 13:917790. doi: 10.3389/fimmu.2022.917790). These findings suggest that low cytokine concentrations may reflect the intrinsic secretory function of hTNSCs rather than being a result of our experimental technique. 

As the reviewer noted, comparing such low concentrations may be of limited significance. However, if a substance is naturally secreted in small amounts, even those minimal quantities may exert functional effects, and slight variations in concentration could be biologically meaningful.

  1. The interpretation in the main text (highlighted in yellow) does not agree with the data presented and needs to be modified.

â—Ž Reply:

There were typos that misled the data understanding. As the comment of reviewer, we corrected the mistakenly written contents.

Comment on Figure 8:

What does the Y-axis represent? This should be explained in the figure legend or indicated on the graph.

â—Ž Reply:

Y-axis represent Relative amounts of gene expression.

According to the comment of reviewer, we added the Y axis meaning.

Round 2

Reviewer 1 Report

Comments and Suggestions for Authors

All critique was addressed.